

# Nonspecific stress response to temperature increase in *Gammarus lacustris* Sars with respect to oxygen-limited thermal tolerance concept

Kseniya Vereshchagina[1,2], Elizaveta Kondrateva[1], Denis Axenov-Gribanov[1,2], Zhanna Shatilina[1,2], Andrey Khomich[3], Daria Bedulina[1], Egor Zadereev[4,5] and Maxim Timofeyev[1]

[1] Institute of Biology, Irkutsk State University, Irkutsk, Russia
[2] Baikal Research Centre, Irkutsk, Russia
[3] International Sakharov Environmental Institute, Belarusian State University, Minsk, Belarus
[4] Institute of Biophysics SB RAS, Krasnoyarsk Research Center SB RAS, Krasnoyarsk, Russia
[5] Siberian Federal University, Krasnoyarsk, Russia

Corresponding author
Maxim Timofeyev,
m.a.timofeyev@gmail.com

## ABSTRACT

The previously undescribed dynamics of the heat shock protein HSP70 and subsequent lipid peroxidation products have been assessed alongside lactate dehydrogenase activity for *Gammarus lacustris* Sars, an amphipod species from the saltwater Lake Shira (Republic of Khakassia). Individuals were exposed to a gradual temperature increase of 1 °C/hour (total exposure duration of 26 hours) starting from the mean annual temperature of their habitat (7 °C) up to 33 °C. A complex of biochemical reactions occurred when saltwater *G. lactustris* was exposed to the gradual changes in temperature. This was characterized by a decrease in lactate dehydrogenase activity and the launching of lipid peroxidation. The HSP70 level did not change significantly during the entire experiment. In agreement with the concept of oxygen-limited thermal tolerance, an accumulation of the most toxic lipid peroxides (triene conjugates and Schiff bases) in phospholipids occurred at the same time and temperature as the accumulation of lactate. The main criterion overriding the temperature threshold was, therefore, the transition to anaerobiosis, confirmed by the elevated lactate levels as observed in our previous associated study, and by the development of cellular stress, which was expressed by an accumulation of lipid peroxidation products. An earlier hypothesis, based on freshwater individuals of the same species, has been confirmed whereby the increased thermotolerance of *G. lacustris* from the saltwater lake was caused by differences in energy metabolism and energy supply of nonspecific cellular stress-response mechanisms. With the development of global climate change, these reactions could be advantageous for saltwater *G. lacustris*. The studied biochemical reactions can be used as biomarkers for the stress status of aquatic organisms when their habitat temperature changes.

## INTRODUCTION

Temperature is one of the factors that determines function and stability of ecosystems. Temperature defines a number of processes in living organisms at all levels of organization (*Iacarella et al., 2015*; *Huey, Buckley & Du, 2018*). In recent years, surface temperature of lakes throughout the world has grown significantly (about 0.34 °C within 10 years) (*O'Reilly et al., 2015*; *Yasuhara & Danovaro, 2016*). Such rapid warming is a drastic signal for the need to study comprehensively the impact of climate change on the status of water ecosystems to assess the fauna vulnerability and adaptive capacity. In addition, this induces the need to develop new methods and tools for environmental protection. Studying thermal tolerance mechanisms and energy metabolism components in aquatic organisms in changing ambient temperature is of essential interest and relevance.

Recently to explain the ecological consequences of climate change, the concept of oxygen-and capacity-limited thermal tolerance (OCLTT) has been used (*Pörtner, 2010*). The key idea of this concept is that there is a limited thermal range, or a life stage, of aerobic performance of the species beyond which the aerobic metabolism is no longer possible. Biochemically these ranges can be detected by the accumulation of end products of anaerobiosis, which is followed by the development of cellular stress and activation of nonspecific cellular stress-response (NCSR) (*Kassahn et al., 2009*).

Among the NCSR components, high attention was given to the antioxidant enzymes (*Almeida et al., 2002*), heat shock proteins, such as HSP70 (*Triebskorn et al., 2002*), lipid and fatty acid composition (*Bergé & Barnathan, 2005*), lipid peroxidation products (*Valavanidis et al., 2006*), enzymes and products involved in energy metabolism including anaerobiosis (*Almeida et al., 2002*), gene expression (*Lee et al., 2008*), etc.

However, it is unknown whether different populations of the same species vary in activation of NCSR on the edge of their oxygen performance range. In our previous studies we investigated inter-populational differences of energy metabolism during gradual warming in two distant populations of the common Holarctic amphipod *Gammarus lacustris Sars, 1863* (*Vereshchagina et al., 2016*) from freshwater and saline lakes, and NCSR capacity of this species from the freshwater reservoir (*Axenov-Gribanov et al., 2016*).

The aim of the present study was to investigate the dynamics of HSP70 and lipid peroxidation products along with activity of lactate dehydrogenase, as biomarker of anaerobiosis, during gradual temperature increase in *Gammarus lacustris* from the saline Lake Shira (Republic of Khakassia, Russia).

Heat shock proteins of the HSP70 family protect and restore the structure of cellular proteins under different stresses (*Mayer & Goloubinoff, 2017*). In many organisms when exposed to stress conditions the amount of HSP70 is elevated due to the increase of damaged proteins (*Axenov-Gribanov et al., 2016*; *Garbuz & Evgen'ev, 2017*). Lactate dehydrogenase catalyzes the reaction of the interconversion of lactate and pyruvate and associated with the processes of carbohydrate and energy metabolism. The enzyme plays an important role in adaptive reactions of the whole organism (*Holbrook et al., 1975*). Lactate dehydrogenase activity depends on such parameters as intensity of swimming and the availability of their food (*Dahlhoff, 2004*). There is a wide range of studies in which this parameter was used

as an indicator of changes in energy metabolism under stress (*Brown-Peterson et al., 2005*). Another indicator studied in our work is the level of lipid peroxidation, determined by the dynamics of the content of its products. Peroxide oxidation processes occur in cell lipids (mostly in membrane phospholipids) as a result of the action of reactive oxygen species (ROS) (*Guéraud et al., 2010*). The processes of peroxidation are series of a chain reactions resulting in a number of products are consistently formed, most of which are toxic (*Valavanidis et al., 2006*).

*G. lacustris* is a suitable model to experimentally investigate the impact of different abiotic and biotic stress factors. It has a wide distribution across Northern Hemisphere (*Wilhelm & Schindler, 2000*). *G. lacustris* inhabits lentic and lotic ecosystems and has a wide ecological valence (*Väinölä et al., 2007*; *Takhteev, Berezina & Sidorov, 2015*). Food spectrum of *G. lacustris* is broad. Being an opportunistic species, in standard conditions it prefers detritus and plant material (*Gladyshev et al., 2000*). From the previous study, preferable temperature for this species is 15–16 °C and it is highly tolerant to a wide range of environment pH variations (6.2–9.2) (*Timofeyev, 2010*). Also, *G. lacustris* is highly tolerant to hypoxia, especially in low water temperature. Thereby, this species is a regular inhabitant of eutrophic water bodies. In addition, this species is an indispensable component of many ecosystems. Thereby, *G. lacustris* is a top predator in the food chain in Lake Shira (Republic of Khakassia). It is noteworthy that the juvenile representatives predominantly inhabit a depth of 1.5–2 m, whereas adult individuals stay apart and live at a depth 5–12 m (*Yemelyanova, Temerova & Degermendzhy, 2002*). Due to the fact, that this species is found in most water bodies of the Holarctic, it can be used as an object of bioindication in assessing the impact of climate change on water bodies and their ecosystems.

## MATERIAL AND METHODS

### Sampling site

*G. lacustris* were caught in July 2013 with a plankton net at depth of 7 m from the southern shore of Lake Shira. The temperature recorded at the time of sampling was 15 °C. The lake is located in Southern Siberia (54°29'7.25"N, 90°12'1.49"E), in the steppe zone of the northern part of the Minusinsk Valley (Republic of Khakassia, Russia). Lake Shira is a brackish meromictic water body with a shape of 9. 35× 5.3 km and water surface area of 35.9 $km^2$. The maximum depth of the lake reaches 24 m, and the average depth is about 11.2 m (*Degermendzhy et al., 2010*; *Rogozin et al., 2017*). The sampling site represents diverse soils that contain gravel, sand, stone, clay and mud; the sublittoral comprises sand with small stones and gray mud; black mud prevails in pelagic zone (*Yemelyanova, Temerova & Degermendzhy, 2002*).

Lake Shira is one of the most saline water bodies (15–17‰) inhabited by *G. lacustris*. Its chemical composition corresponds to the following anion-cation ratio (mg/L): $Cl^-$ - 2100, $Na^+$ - 2880, $K^+$ - 37, $Mg^{2+}$ - 1080, $CO_3^{2-}$ - 174, $Ca^{2+}$ - 51, $SO_4^{2-}$ - 8010, $HCO_3^-$ - 998, and environmental pH is close to 8.7 (*Kalacheva et al., 2002*). In summer, water temperature in the lake littoral zone can reach 28 °C, while the mean annual temperature of water is about 7 °C (*Rogozin et al., 2017*).

## Experimental design and animal maintenance

In this study, experiments were carried out during July 2013 at field station of the Institute of Biophysics SB RAS which is located directly at Lake Shira. Animals were selected with approximately the same size of 8–10 mm. According to the study (*Zadereev & Gubanov, 2002*) this body length can be used to classify adult animals. Immediately after sampling, amphipods were transferred to the laboratory. One hundred individual amphipods were placed into 2 L glass tanks containing aerated 7 °C (i.e., the average temperature of the lake) filtered water from their native habitat. Prior to experimental exposure, amphipods were pre-acclimated for seven days. Tanks with amphipods were kept in a refrigerated showcase (Biryusa, Krasnoyarsk, Russia) to maintain constant temperature during pre-acclimation. Water was exchanged once every two days. The experimental animals were fed daily with potatoes ad libitum. Excess food was removed. During acclimation, the amphipods showed high motor activity and no deaths, which can indicate that the acclimation conditions were not stressing for this species.

Gradual temperature increase experiments were carried out at the rate of 1 °C per hour by use of a refrigerated bath circulator (CRYO-VT-11, Tomsk, Russia) continuing until 100% mortality occurred (modified from *Sokolova & Pörtner (2003)*). After every 2 °C of temperature increase (i.e., every 2 hours) four specimens were randomly collected from each tank, thus, between three to eight replicates were taken at each temperature treatment and shock-frozen in liquid nitrogen. Fixations were conducted upon reaching definite temperatures –9 °C (2 h of exposure), 11 °C (4 h), 13 °C (6 h), 15 °C (8 h), 17 °C (10 h), 19 °C (12 h), 21 °C (14 h), 23 °C (16 h), 25 °C (18 h), 27 °C (20 h), 29 °C (22 h), 31 °C (24 h) and 33 °C (26 h).

## Biochemical methods
### Assessment of heat shock proteins 70 content

Total protein was isolated in 0.1 M Tris HCl (pH 7.6). The amount of protein in samples was determined using the M. Bradford method (*Bradford, 1976*) at 595 nm wavelength. Optical density was measured using the Cary 50 UV/VIS spectrophotometer (Varian Inc., Belrose, Australia). HSP70 dynamics was determined using standard sodium dodecyl sulfate polyacrylamide gel electrophoresis (SDS-PAGE) in 12.5% polyacrylamide gel, followed by Western blotting (*Laemmli, 1970*). For HSP70 visualization, at first, the obtained membranes were incubated with antibodies to HSP70 (monoclonal Anti-Heat Shock Protein 70 antibody produced in mouse; Sigma-Aldrich # H5147, 1:1000 dilution, St. Louis, MO, USA). Then, after washing off the unbound antibody, the membranes were incubated in the solution of secondary antibodies conjugated with alkaline phosphatase (Anti-Mouse IgG (whole molecule)—Alkaline Phosphatase antibody produced in goat, Sigma-Aldrich # A3562, 1:1000 dilution). We used actin as the reference protein. For actin visualization, the following antibodies were used: polyclonal anti-$\alpha$-actin antibodies produced in rabbit (Sigma-Aldrich #A2668, 1:1000 dilution) and secondary anti-rabbit antibodies (Sigma-Aldrich #A9919, 1:1000 dilution). Hsp70 and actin levels were measured by semi-quantitative analysis of grey values on scanned Western blot membranes using

ImageJ Package (v.1.41., Wayne Rasband, NIH, USA). The levels of Hsp70 were normalized relative to $\alpha$-actin expression in each sample and given in arbitrary units (arb. un.).

### Measurement of lactate dehydrogenase activity

Activity of lactate dehydrogenase (LDH) was measured using the enzymatic spectrophotometric method. This method is based on the reaction of pyruvate converting into lactate. NADH to NAD$^+$ oxidation rate is proportional to the LDH activity. Measurements were taken in buffered sodium phosphate solution (0.1 M, pH = 7.5) using the LDH-Vital express kit (B 23.01, Vital–Development Corporation, Saint-Petersburg, Russian Federation) at 340 nm wavelength and $t = 25$ °C according to the manufacturer's instructions. Optical density was measured using the Cary 50 UV/VIS spectrophotometer (Varian Inc., Belrose, Australia).

### Measurement of lipid peroxidation product level

The levels of lipid peroxidation products were estimated from monochromatic light flux absorbed by lipid extract in UV spectrum according to the technique modified from *Deryugina et al. (2010)*. Frozen specimens were ground in 1:1 heptane–isopropanol extraction mixture. Using the extraction mixture, homogenate volume was brought up to 4.5 ml. To separate lipid peroxidase fractions, 1 ml of distilled water was added to samples, which were intensively stirred 10 seconds by hand until fraction became homogeneous then incubated at 25 °C for 30 min. After the phase separation, the isopropanol (lower) and heptane (upper) fractions were centrifuged for 2 min at 14 krpm. 97% ethanol (ratio 1:3) we added to the obtained supernatant, and then the optic density of the solution was measured using the Cary 50 Conc UV/VIS spectrophotometer (Varian Inc., Belrose, Australia). Diene and triene conjugates, and Schiff bases were measured at wave lengths of 232, 278 and 400 nm, respectively. Content of lipid peroxidation products was estimated in arbitrary units (arb. un.) in terms of isolated double bonds as measured at 220 nm wavelength.

### Statistical analysis

All the experiments were carried out with three–eight biological replicates, and biochemical measurements for each sample were performed in triplicate (technical replicates). Immunoblots were analyzed using the ImageJ package (v.1.41., Wayne Rasband, NIH, USA). Normality was checked with the Kolmogorov-Simonov test. Data analysis was performed using the one-way ANOVA test, and the Student-Newman-Keuls test was used as a post hoc-test. When the data distribution deviated from the normal, Kruskal-Wallis with Dunn test as a post hoc-test was used. With $p$-value <0.05, the differences were considered to be significant (to check statistical hypotheses with multiple testing, we also used the Bonferroni correction). Statistical data processing was performed with SigmaPlot package (version 12, Systat Software Inc., USA/Canada).

## RESULTS

In this study, we show that the HSP70 level did not change significantly during the entire exposure to gradual temperature increase from 7 °C to 33 °C (Fig. 1).
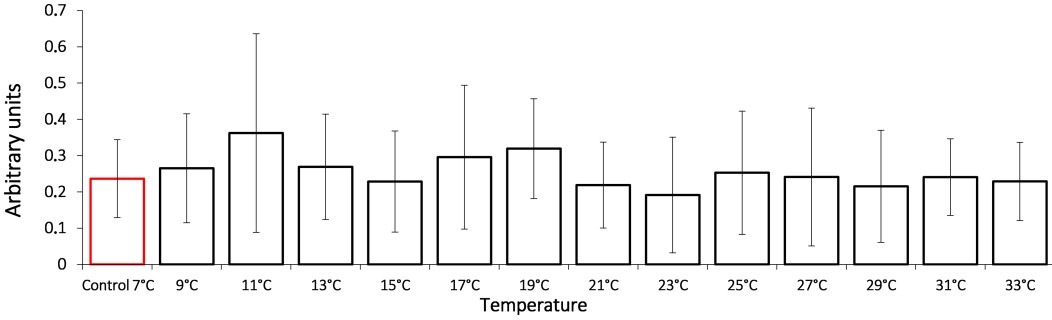

**Figure 1   HSP70 levels in Lake Shira *G. lacustris* amphipods during exposure to gradual temperature increase (1 °C/h).** HSP70 levels presented in arbitrary units. Columns highlighted by red outline indicate the control level. Data are presented as means ± standard deviation of the mean. Ind. –indicates number of individuals of amphipods. Number of replicates: *n*, 7 °C = 3 (12 ind.); *n*, 9 °C = 5 (20 ind.); *n*, 11 °C = 4 (16 ind.); *n*, 13 °C = 3 (12 ind.); *n*, 15 °C = 4 (16 ind.); *n*, 17 °C = 5 (20 ind.); *n*, 19 °C = 4 (16 ind.); *n*, 21 °C = 5 (20 ind.); *n*, 23 °C = 4 (16 ind.); *n*, 25 °C = 4 (16 ind.); *n*, 27 °C = 4 (16 ind.); *n*, 29 °C = 4 (16 ind.); *n*, 31 °C = 5 (20 ind.); *n*, 33 °C = 4 (16 ind.).

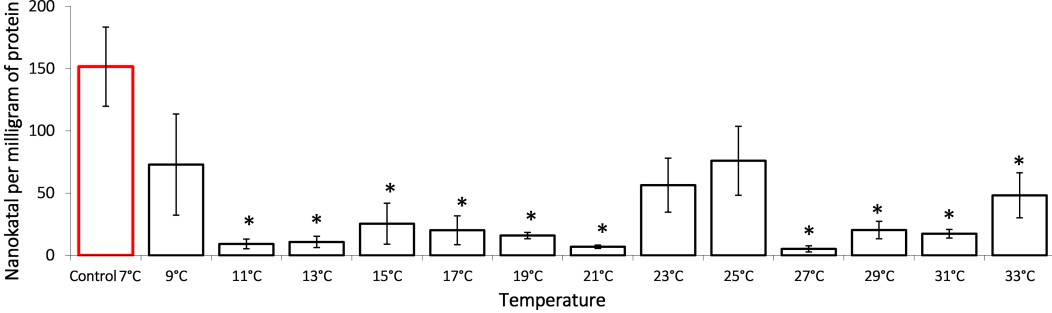

**Figure 2   Lactate dehydrogenase activity (in nKat/mg of protein) in Lake Shira *G. lacustris* amphipods during exposure to gradual temperature increase (1 °C/h).** Columns highlighted by red outline indicate the control level. Asterisks (*) denotes a significant difference ($p < 0.05$) from the control 7°C. Data are presented as means ± standard deviation of the mean. Ind. –indicates number of individuals of amphipods. Number of replicates: n, 7 °C = 3 (12 ind.); n, 9 °C = 5 (20 ind.); *n*, 11 °C = 3 (12 ind.); *n*, 13 °C = 3 (12 ind.); *n*, 15 °C = 4 (16 ind.); *n*, 17 °C = 4 (16 ind.); *n*, 19 °C = 4 (16 ind.); *n*, 21 °C = 3 (12 ind.); *n*, 23 °C = 4 (16 ind.); *n*, 25 °C = 3 (12 ind.); *n*, 27 °C = 4 (16 ind.); *n*, 29 °C = 3 (12 ind.); *n*, 31 °C = 3 (12 ind.); *n*, 33 °C = 4 (16 ind.).

It is shown that the gradual temperature increase leads to lactate dehydrogenase activity decrease, an important component of anaerobic metabolism (Fig. 2). During the exposure, *G. lacustris* demonstrated a reliable 16-fold decrease of the enzyme activity from 151.57 ± 3.80 nKat/mg of protein to 9.19 ± 3.83 nKat/mg of protein on reaching 11 °C. Within the range from 11 °C to 21 °C, the enzyme activity remained low. After that, we observed a short-term reactivation up to 56.37 ± 21.68 nKat/mg of protein and 75.97 ± 27.69 nKat/mg protein in exposure temperatures of 25 °C and 27 °C, respectively. However, at the exposure temperatures over 27 °C, the enzyme activity was again lower than the control levels.

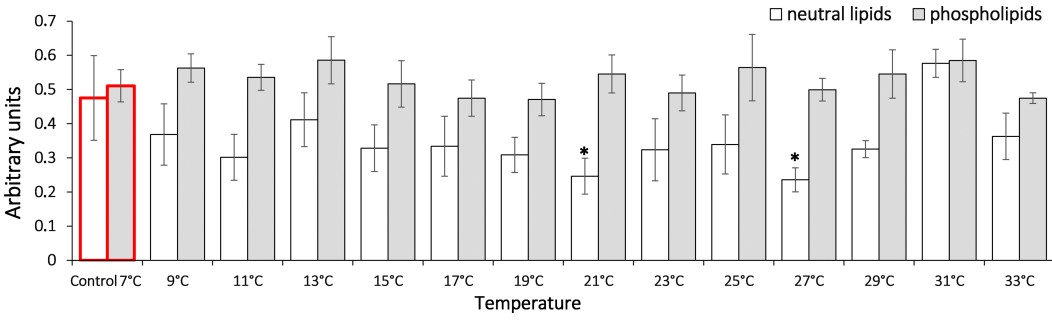

**Figure 3 Levels of diene conjugates in neutral lipids (heptane fraction) and phospholipids (isopropanol fraction) in Lake Shira *G. lacustris* amphipods during exposure to gradual temperature increase (1 °C/h).** Diene conjugate levels presented in arbitrary units. Columns highlighted by red outline indicate the control level. Asterisks (*) denotes a significant difference ($p < 0.05$) from the control 7° C. Data are presented as means ± standard deviation of the mean. Ind. –indicates number of individuals of amphipods. Number of replicates (neutral lipids): n, 7 °C = 7 (28 ind.); *n*, 9 °C = 5 (20 ind.); *n*, 11 °C = 6 (24 ind.); *n*, 13 °C = 6 (24 ind.); *n*, 15 °C = 7 (28 ind.); *n*, 17 °C = 7 (28 ind.); *n*, 19 °C = 6 (24 ind.); *n*, 21 °C = 6 (24 ind.); *n*, 23 °C = 7 (28 ind.); *n*, 25 °C = 5 (20 ind.); *n*, 27 °C = 6 (24 ind.); *n*, 29 °C = 7 (28 ind.); *n*, 31 °C = 7 (28 ind.); *n*, 33 °C = 6 (24 ind.). Number of replicates (phospholipids): n, 7 °C = 8 (32 ind.), n, 9 °C = 7 (28 ind.); *n*, 11° C = 6 (24 ind.); *n*, 13 °C = 6 (24 ind.); *n*, 15 °C = 7 (28 ind.); *n*, 17 °C = 6 (24 ind.); *n*, 19 °C = 6 (24 ind.); *n*, 21 °C = 7 (28 ind.); *n*, 23 °C = 7 (28 ind.); *n*, 25 °C = 6 (24 ind.); *n*, 27 °C = 6 (24 ind.); *n*, 29 °C = 7 (28 ind.); *n*, 31 °C = 7 (28 ind.); *n*, 33 °C = 6 (24 ind.).

To assess the dynamic of oxidation processes under the temperature increase, we measured the content of lipid peroxidation products such as diene conjugates, triene conjugates and Schiff bases. These metabolites reflect various oxidation stages in an organism. Our data shows that in *G. lacustris* decreased the level of diene conjugates in the neutral lipid fraction (Fig. 3) when temperature reached 21 °C (0.24 ± 0.05) and 27 °C (0.24 ± 0.04). Changes of diene conjugates content in phospholipid fraction were not observed until the end of the experiment.

It is shown that the gradual temperature increase led to elevated triene conjugates in *G. lacustris* relative to the control level in composition of both phospholipids and neutral lipids (Fig. 4). Elevated levels of triene conjugates were observed in neutral lipids composition, on reaching 31 °C (two-fold as compared to the basal level, 0.25 ± 0.03). With phospholipids, the elevation occurred on reaching 31 and 33 °C (0.24 ± 0.03 and 0.24 ± 0.02, respectively).

In analysis of the gradual temperature increase effect on lipid peroxidation end products (Fig. 5) in saltwater *G. lacustris*, both fractions showed changes in levels of Schiff bases at 31 °C. It should be noted that in case of neutral lipids, the reaction was short-term, while in the phospholipid fraction, content of Schiff bases deviated from the basal level till the end of exposure.

## DISCUSSION

Various features of the nonspecific cellular stress-response (NCSR) have been outlined here for a saltwater population of *G. lacustris* in order to describe the molecular underpinnings of this species' adaptive potential. This is particularly pertinent, considering that in the near future, it is predicted that this species might be subjected to changing thermal conditions.

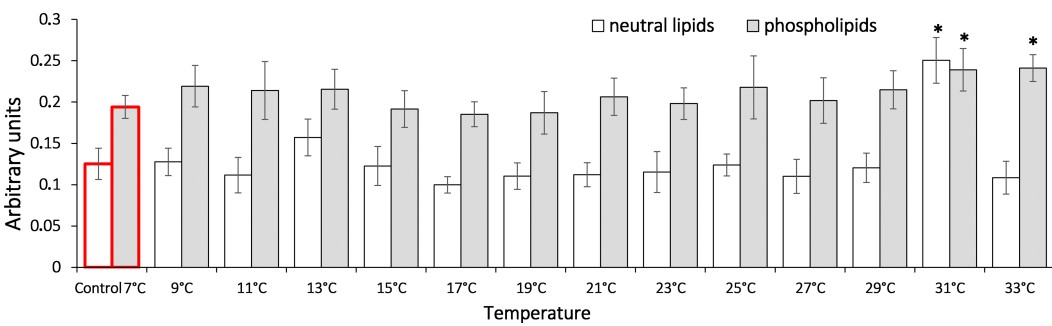

**Figure 4** **Levels of triene conjugates in neutral lipids (heptane fraction) and phospholipids (isopropanol fraction) in Lake Shira *G. lacustris* amphipods during exposure to gradual temperature increase (1 °C/h)** Triene conjugate levels presented in arbitrary units. Columns highlighted by red outline indicate the control level. Asterisks (*) denotes a significant difference ($p < 0.05$) from the control 7 °C. Data are presented as means ± standard deviation of the mean. Ind. –indicates number of individuals of amphipods. Number of replicates (neutral lipids): n, 7 °C = 5 (20 ind.); n, 9 °C = 5 (20 ind.); n, 11 °C = 7 (28 ind.); n, 13 °C = 6 (24 ind.); n, 15 °C = 7 (28 ind.); n, 17 °C = 6 (24 ind.); n, 19 °C = 7 (28 ind.); n, 21 °C = 6 (24 ind.); n, 23 °C = 7 (28 ind.); n, 25 °C = 5 (20 ind.); n, 27 °C = 7 (28 ind.); n, 29 °C = 6 (24 ind.); n, 31 °C = 7 (28 ind.); n, 33 °C = 6 (24 ind.). Number of replicates (phospholipids): n, 7 °C = 8 (32 ind.); n, 9 °C = 7 (28 ind.); n, 11 ° C = 7 (28 ind.); n, 13 °C = 6 (24 ind.); n, 15 °C = 7 (28 ind.); n, 17 °C = 7 (28 ind.); n, 19 °C = 6 (24 ind.); n, 21 °C = 7 (28 ind.); n, 23 °C = 7 (28 ind.); n, 25 °C = 6 (24 ind.); n, 27 °C = 6 (24 ind.); n, 29 °C = 7 (28 ind.); n, 31 °C = 7 (28 ind.); n, 33 °C = 6 (24 ind.).

**Figure 5** **Levels of Schiff's bases in neutral lipids (heptane fraction) and phospholipids (isopropanol fraction) in Lake Shira *G. lacustris* amphipods during exposure to gradual temperature increase (1 °C/h)** Schiff's base levels presented in arbitrary units. Columns highlighted by red outline indicate the control level. Asterisks (*) denotes a significant difference ($p < 0.05$) from the control 7 °C. Data are presented as means ± standard deviation of the mean. Ind. –indicates number of individuals of amphipods. Number of replicates (neutral lipids): n, 7 °C = 7 (28 ind.); n, 9 °C = 6 (24 ind.); n, 11 °C = 6 (24 ind.); n, 13 °C = 5 (20 ind.); n, 15 °C = 7 (28 ind.); n, 17 °C = 6 (24 ind.); n, 19 °C = 7 (28 ind.); n, 21 °C = 6 (24 ind.); n, 23 °C = 7 (28 ind.); n, 25 °C = 6 (24 ind.); n, 27 °C = 7 (28 ind.); n, 29 °C = 5 (20 ind.); n, 31 °C = 7 (28 ind.); n, 33 °C = 6 (24 ind.). Number of replicates (phospholipids): n, 7 °C = 8 (32 ind.); n, 9 °C = 7 (28 ind.); n, 11 °C = 6 (24 ind.); n, 13 °C = 6 (24 ind.); n, 15 °C = 7 (28 ind.); n, 17 °C = 7 (28 ind.); n, 19 °C = 6 (24 ind.); n, 21 °C = 7 (28 ind.); n, 23 °C = 7 (28 ind.); n, 25 °C = 6 (24 ind.); n, 27 °C = 7 (28 ind.); n, 29 °C = 7 (28 ind.); n, 31 °C = 6 (24 ind.); n, 33 °C = 6 (24 ind.).

The saltwater individuals of *G. lacustris* are more thermoresistant than freshwater individuals of the same species. The time taken for 50% of individuals held at 30 °C to reach mortality (LT50) is 15.1 hours less in freshwater specimens than in those from saltwater populations (*Vereshchagina et al., 2016*). Additionally, the thermal tolerance of these saltwater populations is associated with a lower metabolic cost than in freshwater populations. There is a lower energetic demand associated with sustaining osmotic pressure for *G. lacustris* individuals in a salt lake, as the water is isoosmotic to the hemolymph of saltwater amphipods. In contrast, the freshwater populations live in a hypoosmotic environment, which is costly on metabolic energy and affects their ability to provide energy for NCSR (*Vereshchagina et al., 2016*).

The components of NCSR are highly conserved; they are well described in a number of model and non-model organisms (*Lushchak, 2011*; *Elder & Seibel, 2015*). Nevertheless, the molecular basis of adaptation, which are caused by the micro-evolution of NCSR regulatory pathways, are still not described. There are some significant NCSR components, such as heat shock proteins (HSPs) and specifically HSP70, which are essential in cellular protection during environmental change. In terms of evolution, HSPs are highly conserved proteins that are found in all organisms from bacteria to humans (*Rhee et al., 2009*; *Sakharov et al., 2009*; *Xie, 2017*). As molecular chaperones, HSPs participate in multiple cellular processes including protein folding and transport of proteins through membranes; they also take part in renaturation of cellular proteins that were partially denatured by proteotoxic stressors (*Tomanek, 2010*; *Shatilina et al., 2011*). The participation of HSP70 in response to temperature change has been shown in many organisms, where these proteins function as protectors preventing degradation of cellular proteins (*Sørensen, Kristensen & Loeschcke, 2003*; *Timofeyev & Steinberg, 2006*). However, elevation of the stress-induced HSP70 is energetically demanding; the demanding molecular and biochemical adaptation of the organisms to their temperature niches is often implemented through sustaining the pool of HSP70 proteins in the cells in an amount that is sufficient for protecting cellular proteins from damage under varying abiotic parameters (*Bedulina et al., 2013*; *Garbuz & Evgen'ev, 2017*). This is expressed in high constitutively-synthesized levels of HSP70 in cells, and there is no vibrant response to the stress. Basal levels of both constitutive and stress-inducible HSP70 forms can vary significantly in different species and in separate populations of the same species when populations are adapted to different environmental conditions. Our previous study showed that in *G. lacustris* from a freshwater population (Irkutsk region), a significant nine-fold HSP70 elevation was observed under gradual temperature change up to 31 °C (*Axenov-Gribanov et al., 2016*). It is worth noting that multifold HSP70 accumulations in the freshwater population were observed immediately prior to the critical thermal mortality point of 100% individuals (*Axenov-Gribanov et al., 2016*). The results of the current study demonstrate that such multifold elevations of HSP70 levels do not occur in saltwater lake populations of *G. lacustris*. This indicates that there are key differences in the mechanisms for cellular regulation of stress-induced HSP70 synthesis in different populations of the same species. Revealing the nature of such differences is essential for understanding the molecular basis of the phenotypic plasticity in this species when adapting to various environmental changes.
We hypothesize that there are high levels of constitutive HSP70, other molecular chaperones, and other NCSR components in the cells of saltwater animals which may possibly explain the lack of HSP70 accumulation observed here. Our previous study (*Vereshchagina et al., 2016*) found elevated levels of antioxidant enzyme activity for catalase and glutathione S-transferase in saltwater *G. lacustris* when compared to the freshwater population. These high levels suggest higher constitutive levels of NCSR in cells; however further research is required to confirm the causes of the observed differences.

It is known that in ambient temperature variation, the energy deficiency of cells increases due to a malfunction of the electron transport chain, which forces the organisms to activate less efficient energy recovery pathways, particularly in the anaerobic glycolysis pathway, leading to changes the activity of lactate dehydrogenase (*Axenov-Gribanov et al., 2016*). During glycolysis, lactate dehydrogenase catalyses in the reversible reaction of the pyruvate–lactate conversion. In the presence of oxygen, pyruvate converts into acetyl coenzyme A and enters the Krebs cycle; however, in anaerobic conditions, or if the mitochondrial electron transport chain is damaged, pyruvate is reversibly converted into lactate (*Devlin, 2011*). During this study, it was shown that in *G. lacustris*, the activity of lactate dehydrogenase decreased 16-fold in the first stages of the experiment, when the temperature increased from 7 °C to 11 °C (Fig. 2). This decrease in lactate dehydrogenase activity may have been induced by pyruvate levels increasing during glucose oxygen catabolism. In concentrations over 4 mM (*Fregoso-Peñuñuri et al., 2017*), pyruvate is capable of inhibiting the enzyme activity in crustaceans. In another study, we demonstrated the elevation of adenosine triphosphate (ATP) and the depletion of glucose content in the first phases of exposure to gradual temperature increase when starting from 13 °C (*Vereshchagina et al., 2016*). This also denotes the activation of glucose oxygen catabolism. In that same study, at 23 °C, lactate dehydrogenase activity increased up to control levels again, which provided evidence for activation of anaerobic glycolysis processes at these temperatures; though, when 27 °C is reached, the enzyme activity reduced again and remained low until the end of the experiment. It is worth noting that the multifold accumulation of lactate, as the main marker for anaerobiosis in crustaceans, was observed in this population only when reaching 31 °C. At the same time, the adenosine triphosphate level did not show a significant decrease (*Vereshchagina et al., 2016*).

In our early studies, the decreased lactate dehydrogenase activity in gradual temperature increase was also observed in *G. lacustris* freshwater population on reaching 17 °C. Nevertheless, we observed a direct correlation between the elevation of lactate and decrease of ATP levels. This indirectly provides support for the theory that anaerobic processes prevail over aerobiosis in the freshwater population, with incomplete deactivation of the aerobic process. Considerable and multiple lactate accumulations in the freshwater population were noted on reaching 29 °C, which was also accompanied with a decrease of lactate dehydrogenase activity (*Axenov-Gribanov et al., 2016*; *Vereshchagina et al., 2016*). Thus, the data received in this study supports our early results concerning significant differences in energy metabolism regulation between *G. lacustris* salt and freshwater populations.

One of the possible causes of shifting energy balance toward anaerobiosis may be the increasing oxidation processes in cells, and the development of oxidative stress (*Sokolova et al., 2012*). The latter often occurs when an accumulation of oxygen actively forms. In this study, oxidative stress is indicated by the change in the levels of lipid peroxidation products –diene (primary products) and triene (secondary products) conjugates, and Schiff bases (end products) in *G. lacustris* (Figs. 3, 4, 5). It is notable that accumulation of the most toxic lipid peroxides (triene conjugates and Schiff bases) in phospholipids occurs at the same time and temperature of exposure as the accumulation of lactate (*Vereshchagina et al., 2016*). This supports the concept of oxygen-and capacity-limited thermal tolerance (OCLTT) (*Pörtner, Bock & Mark, 2017*). The concept deals with molecular mechanisms sustaining oxygen metabolism, which determine the thermal tolerance limits for each species. According to this concept, when environmental parameters deviate from optimal values, organisms switch their metabolism to anaerobiosis. This leads to an accumulation of lactate and other products of anaerobic metabolism (acetate, succinate etc.) in the tissues of animals. Development of cellular stress, changes in the structure and functions of cell membranes, and the activation of lipid peroxidation processes occur at the same time.

In our study, in *G. lacustris* in neutral lipids (heptane fraction), a decreased level of diene conjugates was observed when the temperature reached 21 °C (Fig. 3). Since no relevant growth was observed in triene conjugate and Schiff base levels, this demonstrates that low-molecular antioxidants are included in antioxidant protection (*Keniya, Lukash & Gus'kov, 1993*; *Mittler, 2002*).

## CONCLUSIONS

Gradual temperature increase caused a complex of biochemical reactions in the saltwater *G. lacustris* studied here, which were expressed by reduced lactate dehydrogenase activity and the activation of lipid peroxidation. There was no multifold increase in HSP70 levels, possibly due to the initially high pool of these proteins in cells, which is energy-efficient for these organisms. The obtained data support the earlier hypothesis that the increased thermotolerance of *G. lacustris* from the saltwater Lake Shira, as compared to a freshwater lake population of the same species, is caused by the differences in energetic metabolic processes and the energy supply of NCSR mechanisms (*Axenov-Gribanov et al., 2016*; *Vereshchagina et al., 2016*). With the development of global climate warming, these reactions could be advantageous for saltwater *G. lacustris*. Additionally, the studied biochemical reactions can be used as biomarkers for the stress status of aquatic organisms when their habitat temperature changes.

## ACKNOWLEDGEMENTS

We express our gratitude to the team of the Laboratory of Biophysics of Ecosystems at the Institute of Biophysics SB RAS for the accommodation and help during Lake Shira field campaigns. Also, we are indeed grateful to Polina Drozdova, Evgenii Protasov and Molly Czachur for proofreading of the article.

### Funding

The study was carried out with the main financial support of Russian Science Foundation grant 17-14-01063, with the partial financial support of Russian Foundation for Basic Research grants 16-34-00687, 16-34-60060, 17-34-50012, the base part of Goszadanie project 6.9654.2017/8.9, joint program of DAAD and Ministry of education and Science M. Lomonosov (6.12735.2018/12.2) and Lake Baikal Foundation (FOB_02-3/05). There was no additional external funding received for this study. The funders had no role in study design, data collection and analysis, decision to publish, or preparation of the manuscript.

### Grant Disclosures

The following grant information was disclosed by the authors:
Russian Science Foundation: 17-14-01063.
Russian Foundation for Basic Research: 16-34-00687, 16-34-60060, 17-34-50012.
Goszadanie project: 6.9654.2017/8.9.
DAAD and Ministry of education and Science: 6.12735.2018/12.2.
Lake Baikal Foundation: FOB_02-3/05.

### Competing Interests

The authors declare that they have no competing interests.

### Author Contributions

- Kseniya Vereshchagina and Denis Axenov-Gribanov conceived and designed the experiments, performed the experiments, analyzed the data, prepared figures and/or tables, authored or reviewed drafts of the paper, approved the final draft.
- Elizaveta Kondrateva performed the experiments, analyzed the data, authored or reviewed drafts of the paper, approved the final draft, performed sampling of invertebrates and biochemical measurements.
- Zhanna Shatilina conceived and designed the experiments, analyzed the data, authored or reviewed drafts of the paper, approved the final draft.
- Andrey Khomich performed the experiments, analyzed the data, authored or reviewed drafts of the paper, approved the final draft, performed sampling of invertebrates, provided technical support.
- Daria Bedulina analyzed the data, authored or reviewed drafts of the paper, approved the final draft.
- Egor Zadereev conceived and designed the experiments, analyzed the data, contributed reagents/materials/analysis tools, authored or reviewed drafts of the paper, approved the final draft.
- Maxim Timofeyev conceived and designed the experiments, analyzed the data, contributed reagents/materials/analysis tools, prepared figures and/or tables, authored or reviewed drafts of the paper, approved the final draft.

## Data Availability

The raw data are provided as Supplemental Files.

## Supplemental Information

Supplemental information for this article can be found online at http://dx.doi.org/10.7717/peerj.5571#supplemental-information.

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
