# Peer review of "Nonspecific stress response to temperature increase in Gammarus lacustris Sars with respect to oxygen-limited thermal tolerance concept"

_PeerJ, doi:10.7717/peerj.5571_

## Round 0.1 · original submission · Major Revisions

I appreciate enormously your interest in PeerJ. Although the study is suitable for publication some major changes are required before the article could be accepted. Please, pay special attention to the reviewer#2' comments. The English language need to be checked deeply in order to improve the quality and understanding of the manuscript.

If you are prepared to undertake these revisions, we look forward to receiving your revised manuscript, and ask that you please directly respond to the readers' comments when you submit it.

Best regards,

Salva

Reviewer 1 ·

Basic reporting

The investigation is a clear contribution to knowledge on the impact of climate change, which is also included in the guidelines of the journal. The expression of the text is simple, concise and scientific. The research has been carried out with ethics, especially taking into account the use of other species of minor evolutionary scale.

The introduction needs more detail. I suggest to improve the defense of the use of the biomarker and its description, in addition, it should specify other factors that influence even if they will not be analyzed at a later time and why he chose precisely the factor temperature for the study, the goals are not ben defined although the objectives and finally would have to endorse this type of studies with a database of legislation (EPA UNITED NATIONS OECD...) to provide additional justification.
In discussion would have to introduce the application of the biosensor shallow way

Specific Comments

64 to 74 lines Within the methodology it is convenient to remove them and carry the information of the biomarker to introduction
(materials must refer exclusively to the objects used to perform the job, only put the animals that are used, its origin and type of sampling)
66 line: enter if the zooplankton or benthic species is
94 line: why this age , this age are more sensitive?
96-97-98 lines: Why it chose this range of temperatutas
184 line: Appoint Representative species

Experimental design

The research question is well determined, the experimental design is appropriate and modern, however in the statistical analysis, the sample of 3 individuals is low and the repetition of the bioassay should be adjusted to a minimum of n= 6-8 for a good statistic

Validity of the findings

The work complies with the objectives and in any case with the premises, assumptions and arguments raised, the results are clear, concise and sufficient appropriate measurements

Additional comments

The article has good quality but requires certain minor changes to improve the introduction
The article are not defined the goals, although if the objectives
Check the match or contrast with more papers published by other authors even comparing with other bioindicators.
In addition to the conclusions derived directly from the conduct of the investigation should include any alternative interpretation that is discarded during the work.
It should reflect more closely the innovations found.

Reviewer 2 ·

Basic reporting

The manuscript is well structured and conforms to PeerJ standards. The English language needs to be improved to ensure understanding of your text and findings. I have given line item suggestions in general comments below for the early sections of the manuscript. Grammar and sentence structure are an issue throughout and especially in the discussion. Please revise with this in mind, paying special attention to the discussion.

Line 23: Although you are indeed the first to assess the combination, anyone working on non model organisms can use a similar statement as the leading sentence in almost every study they do. I suggest modifying the organization of the introduction to draw in the reader (and also refocus the abstract). Are the amphipods themselves the unique and interesting thing that lead to this study, or was it driven by the question and they are an excellent study organism to address it? You could possibly lead the introduction with the paragraph at line 52 which is well written and indicates your research was question driven.

Figures are relevant and high quality. The figures could be improved by adding more details to the legends. Please include number of replicates and number of amphipods per replicate in figure legends: n for each bar and number of individual amphipods for each n.

I am glad to see the raw data is provided, however I think further organization would clarify how the study was conducted. The western blot gels do not provide what the ladder is to conclude the bands on the gel are actin and HSP70. Also the replicate structure is unclear (see additional questions pertaining to this in experimental design comments). Are the HSP 70 replicates additional measurements on the same initial samples, or repeat measurements that consist of new specimens from the same temperature treatment? No repeat measurements were done on the same samples for LDH, and lipids, but there were for HSP70, is that correct? Is what you call "replicates" in the HSP70 supplement repeat measurements of the same initial samples? My understanding is the Glk022s1 etc are your sample naming system, please label as such. What are the numbers 1-14 etc under the column with no heading in the HSP70 supplement? Why is there no background etc listed for the first HSP70 samples, but there are for all replicates? What are the numbers below the Glk022s1 numbers in the first block of HSP70 raw data?

Experimental design

This research is original primary research and within the scope of the Journal.

The research has been preformed to high technical and ethical standards. When the replicate and repeat measurements structure is better described it will become clear how rigorous the investigation is. Were the same samples used for all biochemical analysis, or unique organisms for each analysis? Based on the numbering in the supplement it appears that unique samples were used for each biochemical analysis, but that needs to be described more clearly.

The Methods description needs to be improved in several key areas to ensure sufficient detail is provided. How many individual organisms were in each 2L aquarium? How was the temperature of the aquarium controlled? Exactly what are the control treatments? Were those organisms kept at 7°C for the same amount of time as the longest temperature experiment? Please justify the 1 degree per hour increase. It is referred to as "gradual", but how does that compare to how fast the environmental temperature changes for this species in the location you collected it? Also it is unclear what you mean by the phrase "ambient temperature". Is that the acclimation temperature in the laboratory? If so ambient is not an accurate description of it. It is stated in the abstract that 7°C is the mean annual temperature of the lake, but what was the temperature of the water when specimens were collected? Did you check the concentrations of all the biochemical markers you examined from specimens frozen in liquid nitrogen directly after collection? This would inform what there constitutive levels are. You state at line 292 that there is a "initially high pool of these proteins in cells" but it may be that HSP 70 was induced in response to the temperature change between environmental temperature and laboratory.


For all biochemical methods: HSP, LDH, lipid peroxidase: how many specimens were used for each measurement? Did you pool specimens, or are measurements on individuals? For example line 102- what you mean by samples? Is that a single individual from each temp treatment? I see in line 97 you say 2-3 specimens per sample. So only 2-3 individuals were exposed to each treatment? Were those 2-3 then combined for one measurement? At line 140 you state all experiments were carried out with 3-8 biological replicates- what does this mean?

Validity of the findings

It is difficult to assess the validity of the findings in the current version of the manuscript. This is due to the mentioned gaps in information on the experimental design, and grammatical and structural errors in the English language throughout the discussion. These issues need to be addressed before the findings can be properly considered.

Additional comments

Line 59 Poor readability. This sentence needs to be improved to ensure the research question is well defined, and the phrase " ambient temperature gradual increase" needs to be revised throughout the manuscript (for example also at line 153). It is not clear what you mean by ambient, and this phrase is awkward and too vague.

Line 61: This species is from a saltwater lake, but in line 65 you say lentic an steam ecosystems, implying freshwater. Does it occur in both fresh and salt water habitats? Line 69 you list as euryhaline. I suggest you move the sentence at line 69 up and either combine it with the one at line 65, or have it just before to clarify the habitat these amphipods live in.

Line 65: Add with: "species with a..."

Line 74: "plant food"- change to plant material

Line 84: Please clarify: the plankton tow was done bottom to surface, but you then list above 7 meters depth, do you mean 7 meters water depth? Or do you mean 7m to surface for the tows?

Line 85: Do you have data for temperature range in the natural habitat? Including daily temp fluctuation in the summer at this location, and day night change?

Line 96: Please add what the longest experimental duration was in hours.

Line 98: Why this reference? Are you citing that paper because you fixed in liquid nitrogen in a way suggested in that publication? That citation does not seem relevant here.

Line 112: Please check this antibody number, as the sigma produce #A266 is not for β-actin

Line 122: Please provide product number for LDH express kit, as I could not find it when searching the Vital-Diagnostics webpage.

Line 129-131: Poor wording, restructure as "To separate lipid peroxidase fractions, 1 ml of distilled water was added to samples, which were stirred by xx then incubated at 25°C for 30 min." Please also add how stirring was achieved (xx), and if you want to use intense, explain what you mean by that.

Line 152: Sentence structure issue: why is ambient in the sentence (similar issue with line 59 of the introduction)? You haven't stated what the ambient environmental temperature was when you collected specimens. Do you mean the acclimation temperature (laboratory)? Possibly change to "... during the entire exposure of acclimation at 7°C and gradual (one degree per hour) temperature increase to 33°c "

Line 153: What effect are you refereeing to it imposing? Inhibitory?

Lines 163-164: Sentence structure, try taking out the "in" after "obtained", and add "the" after "conjugates in".

Line 166: Change from till to until, which is formal English.

Lines 178-183: Poorly structured sentence, please revise.

Line 184: Is this according to early experimental data in this study? If so reference where that data is provided. if it is refereeing to data from another study please cite.

Line 191: What do you mean by earlier? Another study? If so please reference it.

Line 203: Provide a reference here.

Line 203: Please change "evidences" to "demonstrates."

Line 242: Change "participates" to "catalyzes".

Line 249: Add s at the end of "concentration".

---

## Round 0.2 · Minor Revisions

I again appreciate enormously your effort in reviewing the manuscript. Although the study substantially improved, some minor changes are required before the article would be accepted. Please, pay special attention to the reviewer#2' comments.

If you are prepared to undertake these revisions, we look forward to receiving your revised manuscript, and ask that you please directly respond to the readers' comments when you submit it.

Best regards,

Salva

Reviewer 1 ·

Basic reporting

'no comment'

Experimental design

'no comment'

Validity of the findings

'no comment'

Additional comments

The n should be specified specifically, not an interval
Should discuss with other bioindicators if there are studies related to the topic and otherwise expose it clearly in the discussion

Reviewer 2 ·

Basic reporting

There are still a few areas that need some clarification. Overall the basic reporting has been improved. I appreciate the author's effort reporting sample size, and number of individuals per replicate and for improving the raw data tables. With some more minor revisions the manuscript should be ready for publication.

Experimental design

The experimental design is clearly explained and justified. My only concern is in the western blot images for HSP 70: the ladder you provided information for has a 75 kda band at the first pink band. The images of the western blot appear to have that band at the top. Could you please mark the Kda band sizes for your ladder to demonstrate that the bands from the samples are at the correct size to interpret as HSP 70. Also you do not refer to a positive control in your methods though it is flagged on the gel. Is actin, what you call your "reference protein" your control? Please clarify and use a consistent term.

Validity of the findings

With some minor revisions to the discussion and conclusions the validity of the findings will be more evident. The data is robust, though the sample size is small due to limitations working on non-model organisms. Conclusions are generally well stated, and supported by the results. Although it is not clear to me that you demonstrate evidence for high constitutive levels of HSP 70. Please clarify, and include references to previous work if that is how you are interpreting this as high.

Additional comments

Line 34 and 35: But you didn’t measure lactate accumulation, only LDH activity. Please revise to either say LDH, or to clarify you are citing a previous study.
Line 74: Change “in gradual temperature” to “during gradual temperature…”
Line 77 Awkward sentence: maybe: In many organisms when exposed to stress conditions the amount of HSP 70 is elevated due to the increase in damaged proteins”
Line 140: Is this your biological replicate you refer to in the statistical analysis section? Need to clarify both sections.
Line 169 “optical density…” needs to be moved up to HSP section. Please provide spectrophotometer information in LDH section. What temperature was it measured at?
Line 227 on: This sentence needs to be rewritten, it doesn’t make sense, “will inhabit” what?
Line 236 Please change “freshwater” to “saltwater”
Line 239 Change “conservative” to “conserved”. “diply” maybe you meant in-depth?
Line 241: What do you mean by “molecular underpinnings”? Rephrase or clarify
Sentences from Line 242 -260, HSP section is repetitive, and should be rewritten to be more concise.
Line 274 Are you saying you consider the levels of constitutive HSP 70 high in this study? I do not see evidence for this, so please clarify and support this. (Also line 337)
Line 295 How instantaneously is LDH synthesized? Would an hour be fast enough for it to increase?
Line 324: “Increase the number” does not make sense to me. Clarify if you mean increase the use of anaerobic pathways, or increase to using more than one?
Line 331 change “this can evidence” to “this demonstrates”
Line 337: change “probably” to “possibly”. You will need to add more to the discussion to justify that you think there is high constitutive levels

---

## Round 0.3 · accepted · Accept

I have checked the changes and I think your manuscript is ready to be accepted in its present form. Congratulations for your effort and work.

Sincerely,

Salva

#